# Multi-Criteria Feature Selection Based Intrusion Detection for Internet of Things Big Data

**DOI:** 10.3390/s23177434

**Published:** 2023-08-25

**Authors:** Jie Wang, Xuanrui Xiong, Gaosheng Chen, Ruiqi Ouyang, Yunli Gao, Osama Alfarraj

**Affiliations:** 1School of Communication and Information Engineering, Chongqing University of Posts and Telecommunications, Chongqing 400065, China; s210131230@stu.cqupt.edu.cn (J.W.); s170101078@stu.cqupt.edu.cn (G.C.); s210131183@stu.cqupt.edu.cn (R.O.); 2School of Software, Dalian University of Technology, Dalian 116024, China; gylllll@mail.dlut.edu.cn; 3Computer Science Department, Community College, King Saud University, Riyadh 11437, Saudi Arabia; oalfarraj@ksu.edu.sa

**Keywords:** internet of things security, intrusion detection, big data, smart cities, feature selection

## Abstract

The rapid growth of the Internet of Things (IoT) and big data has raised security concerns. Protecting IoT big data from attacks is crucial. Detecting real-time network attacks efficiently is challenging, especially in the resource-limited IoT setting. To enhance IoT security, intrusion detection systems using traffic features have emerged. However, these face difficulties due to varied traffic feature formats, hindering fast and accurate detection model training. To tackle accuracy issues caused by irrelevant features, a new model, LVW-MECO (LVW enhanced with multiple evaluation criteria), is introduced. It uses the LVW (Las Vegas Wrapper) algorithm with multiple evaluation criteria to identify pertinent features from IoT network data, boosting intrusion detection precision. Experimental results confirm its efficacy in addressing IoT security problems. LVW-MECO enhances intrusion detection performance and safeguards IoT data integrity, promoting a more secure IoT environment.

## 1. Introduction

IoT big data assumes a pivotal role within the evolving IoT landscape, emerging as a novel network resulting from advancements in Internet technology, catering to diverse user needs. Through the connectivity of myriad objects and entities via sensors such as RFID, GPS, laser scanners, and infrared sensors, the IoT grants them Internet access via predefined network communication protocols. This integration seeks to establish an intelligent network that interconnects and facilitates communication among all entities, seamlessly amalgamating monitoring, identification, management, and localization into a cohesive system. Embedded computing devices serve as the bedrock of IoT, bridging the physical environment with the Internet [1]. Nevertheless, with the widespread adoption and exponential growth of IoT devices, concerns about information security have surged dramatically. Instances of network disruptions and the compromise of sensitive data have become increasingly frequent, with various viruses and attacks posing substantial threats to individuals, businesses, and society at large, leading to considerable economic losses and potential hazards. Consequently, network security has become a pressing and paramount concern requiring immediate attention [2]. Intrusion detection technology emerges as a critical security mechanism, enabling the identification of illicit activities before attackers infiltrate the network. Intrusion detection systems effectively furnish defensive capabilities to shield networks from attacks [3,4]. By employing intrusion detection systems, potential threats can be proactively detected, thus fortifying the overall security of the IoT environment. It is essential to recognize the pivotal role of IoT Big Data in shaping the IoT landscape. The ensuing security challenges underscore the necessity of integrating robust intrusion detection technologies to mitigate risks and guarantee the integrity and confidentiality of IoT systems and the generated data.

Machine learning methods, an interdisciplinary field encompassing computer theory, probability theory, and statistics, have found extensive utility in network intrusion detection. Most classification learning techniques presuppose balanced quantities of training samples for each class. However, owing to substantial dissimilarities in the frequencies of diverse network attacks and other factors, network intrusion data frequently manifest class imbalance, where sample counts diverge significantly across classes. This class imbalance quandary can result in suboptimal classification performance for minority class samples. Consequently, the primary objective of this paper is to address the class imbalance predicament, thereby enhancing the recall of the minority class within network intrusion detection [5]. Additionally, network intrusion data exhibit high dimensionality, wherein only a subset of features holds relevance for sample classification, with others being redundant or inconsequential. The presence of redundant and irrelevant features in high-dimensional data can give rise to reduced accuracy, detection rates, and elevated false alarm rates within network intrusion detection models. To surmount this challenge, the paper introduces the LVW-MECO feature selection algorithm. The primary contributions and innovations of this study are delineated as follows:This paper introduces the HSACEC hybrid sampling algorithm designed to acquire a balanced dataset, effectively tackling the issue of excessive discarding of majority class samples inherent in conventional undersampling methods. Such methods rely solely on the average classification error rate within clusters, particularly when there exists a substantial imbalance between the count of majority class samples and minority class samples.An improved LVW algorithm, called the M-LVW feature selection algorithm, is proposed in this study. This algorithm takes the evaluation criterion of the feature subset as an input parameter, which represents a performance evaluation metric of the classifier. The parameter can be set based on specific requirements. Subsequently, this paper extends the M-LVW algorithm to feature selection in the OVO framework and introduces the LVW-MECO algorithm. Firstly, the LVW-MECO algorithm applies the M-LVW algorithm to each individual base classifier in the OVO scheme. It conducts wrapper-based feature selection using the accuracy of the base classifier as the evaluation criterion for the feature subset, aiming to identify distinct feature subsets for each base classifier.This paper integrates the LVW-MECO algorithm with the BP neural network to establish an LVW-MECO intrusion detection model (LVW-MECO-IDM) for network intrusion detection. Experimental evaluations were conducted on the publicly available IoT-23 network intrusion detection dataset to validate the superiority of the LVW-MECO algorithm. The results demonstrate that the LVW-MECO-IDM, which utilizes the LVW-MECO algorithm, can effectively improve the accuracy and detection rate, and reduce false alarm rates.

The remaining structure of this paper is as follows: Section 2 provides a comprehensive review of the research status of intrusion detection techniques based on feature selection. Section 3 covers the relevant knowledge and technologies. Section 4 addresses the problem of decreased accuracy in network intrusion detection caused by redundant and irrelevant features. It proposes the LVW-MECO feature selection algorithm and the HSACEC hybrid sampling algorithm. Section 5 focuses on the analysis of experimental results. Through experiments, the superiority of the LVW-MECO algorithm is validated. Section 6 provides the conclusion and outlook. 

## 2. Related Work

Feature selection is a crucial step in machine learning as it helps in selecting the most important features for subsequent tasks. Effective feature selection aids in dimensionality reduction, improves prediction accuracy, and enhances the interpretability of results. Many researchers have focused on studying feature selection in the context of network intrusion detection. In reference [6], a multi-agent reinforcement learning framework is proposed to address the feature selection problem. Specifically, they redefine feature selection using a reinforcement learning [7] framework, treating each feature as an agent. Furthermore, three methods, namely statistical description, autoencoders, and graph convolutional networks (GCN), are used to obtain the environmental state, which is then transformed into a fixed-length representation as an input for reinforcement learning. Reference [8] presents a multi-objective feature selection method based on the NSGA-II (Non-dominated Sorting Genetic Algorithm). This method utilizes the Jaccard coefficient as a measure of variable information asymmetry for feature selection. A neural network classifier is then built, leading to an improvement in classification accuracy. In reference [9], a feature selection algorithm based on the squid optimization algorithm is proposed, using a weighted sum of false alarm rate and detection rate as the fitness function. The algorithm is combined with a decision tree algorithm to construct a network intrusion detection model [10], which achieves favorable classification performance.

Reference [11] proposes a feature selection algorithm that combines K-means [12] clustering accuracy, used as a loss function for selecting feature subsets, with a local search algorithm. This algorithm is integrated with a multilayer perceptron to construct a network intrusion detection model, which improves the detection accuracy of the network intrusion detection model. Reference [13] employs a deep learning belief network to identify important features, followed by the adoption of a support vector machine algorithm to establish a classifier, enhancing the performance of network intrusion detection. Reference [14] conducts feature selection based on correlation and information gain, and then builds a network intrusion detection classifier using an artificial neural network. Reference [15] presents a novel network intrusion detection model that combines a logistic regression algorithm with a genetic algorithm-based feature search for feature selection. Reference [16] introduces four novel feature quality metrics and utilizes these metrics to dynamically select useful features during the combination process. The literature [17] proposes a novel machine learning [18] feature selection method, called Unsupervised Discriminative Projection for Feature Selection (UDPFS) to select discriminative features by conducting fuzziness learning and sparse learning, simultaneously. In the literature [19], the Multiple Feature Extraction Extreme Learning Machine (MFE-ELM) algorithm is employed for cloud computing, adding a multi-feature extraction process to cloud servers and using the MFE-ELM algorithm deployed on cloud nodes to detect and discover network intrusions on the cloud nodes. The literature [20] proposes a new Unsupervised Adaptive Feature Selection with Binary Hashing (UAFS-BH) model, which learns binary hash codes as weakly supervised multi-labels and simultaneously exploits the learned labels to guide feature selection. The literature [21] proposes a fully parallelizable feature selection technique intended for the K-means algorithm. The proposal is based on a novel feature relevance measure that is closely related to the K-means error of a given clustering.

## 3. Preliminary

### 3.1. Intrusion Detection Mechanism

Network intrusion refers to network activities that pose a threat to the integrity, confidentiality, and availability of network system resources. With the rapid development of IoT, data security of systems [22] has constantly faced threats from network attacks, and network intrusion detection is an integral part of network security. A network intrusion detection system consists of the following three modules [23]:IoT Information Collection Module: It collects intrusion data and performs statistical analysis based on the feature space of network intrusion detection.Network Intrusion Detection Module: This module uses the data collected by the information collection module to train intrusion detection algorithms. It then uses the model to detect whether the network data are normal. If it is abnormal, it further identifies the type of attack to which the network data belongs.Response Module: This module responds accordingly to the detection results. If an attack is detected, it takes appropriate interception and handling measures.

Network intrusion detection methods can mainly be classified into two categories: misuse-based network intrusion detection methods and anomaly-based network intrusion detection methods [24]. Classification algorithms are applied in network intrusion detection to distinguish between abnormal data and normal data. This approach can be used in both misuse-based and anomaly-based network intrusion detection.

#### 3.1.1. Intrusion Detection Based on Misuse

Intrusion detection methods based on misuse typically rely on a database containing various network attack behaviors to determine whether a detected behavior matches a known attack pattern. Once a matching attack type is identified, the intrusion detection system reports the anomaly, enabling the intrusion prevention system to take appropriate actions promptly. This is the most commonly used detection method, as it is easy to implement in existing network topologies and demonstrates high accuracy in detecting known attacks [25].

However, this detection method can only identify known attack types included in the database. In other words, its drawback lies in its inability to recognize unknown attack types, leading to a relatively high false-negative rate.

#### 3.1.2. Anomaly-Based Network Intrusion Detection

Unlike misuse-based network intrusion detection methods, anomaly-based network intrusion detection methods establish models of normal network behavior and detect intrusion behaviors based on whether the detected data significantly deviates from the normal behavior model. The main advantage of anomaly-based network intrusion detection lies in its ability to detect previously unseen novel attacks. Additionally, anomaly-based network intrusion detection methods perform detection tasks faster than misuse-based methods [26].

However, in reality, such deviations can represent either intrusive behavior or normal behavior that should be added to the model. Therefore, the drawback of anomaly-based network intrusion detection is a high false-positive rate.

#### 3.1.3. Edge Computing-Based Intrusion Detection

Edge computing-based intrusion detection in IoT [27,28,29] is a method that involves traffic monitoring and anomaly detection at the edge nodes of IoT devices [30]. Traditional IoT traffic monitoring typically involves sending all device data to the cloud for processing and analysis, which can lead to high latency, significant network bandwidth consumption, as well as privacy and security concerns [31]. By performing anomaly detection on the edge nodes of the IoT, the amount of data transmitted to the cloud can be reduced, thereby reducing latency and improving response speed.

However, edge devices often have limited computing resources, such as processing power, storage capacity, and memory size. This limitation restricts the ability to perform complex traffic analysis and anomaly detection algorithms on edge nodes [32].

### 3.2. Feature Selection Methods

Feature selection involves two stages: a feature subset search and an evaluation of the feature subset. Based on these two stages, feature selection algorithms can be classified.

#### 3.2.1. Feature Selection Based Search Strategy

Given a feature set with n features, the feature subset has $2^{n}-1$ subsets, which form the feature space for searching the optimal feature subset. The search methods used in feature selection algorithms are known as search strategies [33]. Currently, based on the various search strategies used in feature selection algorithms, they can be classified into three categories: feature selection methods based on a global optimal search strategy, feature selection methods based on a sequential search strategy, and feature selection methods based on a random search strategy, as shown in Figure 1.

#### 3.2.2. Feature Selection Based on Sequential Search Strategy

Feature selection methods based on a sequential search strategy can be categorized into the following four types based on the differences in the starting point and search direction:Feature selection methods based on individual best feature search strategy: In this category of algorithms, the criterion values are first calculated for each feature used individually. Based on these criterion values, the features are sorted, and the top l features are selected as the output feature subset.Feature selection methods based on sequential forward search strategy: These algorithms use a “bottom-up” search approach. Initially, the target feature subset is initialized as an empty set. In each step, the feature that optimizes the evaluation criterion the most is added to the target feature subset. The search ends when the termination condition is met, and the obtained target feature subset is considered the selection result.Feature selection methods based on sequential backward search strategy: These algorithms use a “top-down” search approach. The target feature subset is initialized with all the features, and in each step, an irrelevant feature is removed until the termination condition is satisfied.Feature selection methods based on bidirectional search strategy: These algorithms simultaneously add relevant features and remove irrelevant features in each step.

Feature selection methods based on sequential search strategy have low time complexity [34] and are widely used in practical applications. However, a drawback of this approach is that once a feature is selected or removed during the search process, it cannot be undone, which can lead to obtaining local optima.

#### 3.2.3. Feature Selection Based on Random Search Strategy

This feature selection method utilizes a random search strategy, which has the potential to escape local optima and find approximate optimal solutions. Therefore, in general, feature subsets selected by feature selection methods based on a random search strategy tend to outperform those based on a sequential search strategy.

## 4. Algorithm Design

### 4.1. LVW Algorithm

The LVW algorithm belongs to wrapper-based feature selection methods. It uses a random strategy for the subset search, and the evaluation criterion for the feature subset is the classification error rate of a classifier.

The LVW algorithm is described in Algorithm 1. In the 5th row of the table, the return value of the ‘xx’ function represents the error rate of the classifier h on the feature subset $\mathrm{CrossValidation}\left(h\left(F^{\prime },D\right)\right)$ obtained using cross-validation on the dataset D. If this error rate is lower than the error rate of the current best feature subset $F^{*}$, or if the error rates are the same but $F^{\prime }$ contains fewer features, then $F^{\prime }$ is considered the new best feature subset. The algorithm terminates and outputs the best feature subset $F^{*}$ when there have been no improvements in the feature subset for T consecutive iterations.
**Algorithm 1:** LVW feature selection algorithm**Input:** Given a dataset D, feature set F, classifier algorithm h, and stop condition control parameter T. Process:1. **Initialize:**
err=∞; d=F; F*=F; t=02. **while**
t<T **do**3.   Randomly generate a feature subset F′4.   d′=F′5.   err′=CrossValidationhF′,D,e6.   iferr′<err OR  err′==err AND d′<dthen7.      F*=F′; t=0; err=err′; d=d′8.    **else**9.       t=t+1
10.   **end if**11. **end while****Output:** feature subset F*

### 4.2. OVO Decomposition Strategy

The OVO decomposition strategy is designed for multi-class classification tasks by splitting them into multiple binary classification sub-tasks. The idea is to pair each two classes out of N classes and design a binary classifier, resulting in a total of N(N − 1)/2 binary classifiers. Then, a certain aggregation strategy is used to combine all the binary classifiers into a multi-class classifier. This way, the complex multi-class classification task is broken down into several easier-to-recognize binary classification sub-tasks. The binary classifiers in OVO are also known as base classifiers. To identify the class of unknown samples, the OVO method generally involves the following two steps:1.Each binary classifier returns a pair of confidences $l_{ij},l_{ji}\in \left[0,1\right]$ for an unknown sample, indicating the probability of the unknown sample being classified as class $C_{i}$ relative to class $C_{j}$. Moreover, $l_{ji}=1-l_{ij}$. If the classifier provides only one confidence ll, the other confidence can be calculated based on $l_{ij}$. All the confidences returned by the binary classifiers are combined to form a scoring matrix L:(1)$$L=\left(\begin{array}{cccc} - & l_{12} & \cdots & l_{1N}\\ l_{21} & - & \cdots & l_{2N}\\ \vdots & & & \vdots \\ l_{N1} & l_{N1} & \cdots & - \end{array}\right)$$2.Finally, a certain aggregation strategy is adopted to integrate the outputs of all base classifiers and obtain the predicted class for the unknown sample. Several commonly used aggregation strategies in OVO are as follows:
Voting strategy (VOTE) [35]: This method uses a voting strategy to obtain the final class label by selecting the class with the most votes from all base classifiers. The predicted class label is the output result:
(2)$$H=\underset{i=1,\ldots ,N}{\arg\max}\sum_{1\leq j\leq N,j\neq i}g_{ij}$$In the equation, if $g_{ij}$ is 1, it indicates that the base classifier predicts the unknown sample as class i. If it is 0, it means it does not predict it as class i, and there is:(3)$$g_{ij}=\left\{\begin{array}{ll} 1,\; & l_{ij}>l_{ji}\\ 0,\; & \mathrm{other} \end{array}\right.$$Learning Valued Preference for Classification (LVPC) [36]: This method introduces a conflict level, absolute preference, and unknown degree into the recognition process of the final class. Its decision rule is as follows:
(4)$$H=\underset{i=1,\ldots ,N}{\arg\max}\sum_{1\leq j\leq N,j\neq i}c_{ij}+\frac{1}{2}p_{ij}+\frac{n_{i}}{n_{i}+n_{j}}I_{ij}$$In the equation, $c_{ij}$ and $c_{ji}$ represent the absolute preferences for class i and class j, respectively. $p_{ij}$ represents the level of conflict, $n_{i}$ represents the number of samples for class i in the training set, and $I_{ij}$ represents the unknown degree. The corresponding calculation methods are shown as follows:(5)$$c_{ij}=l_{ij}-\min\left\{l_{ij},l_{ji}\right\}$$
(6)$$p_{ij}=\min\left\{l_{ij},l_{ji}\right\}$$
(7)$$I_{ij}=1-\max\left\{l_{ij},l_{ji}\right\}$$Preference Relations Solved by Non-dominance Criterion (ND) [37]: This method incorporates normalized fuzzy preference relations into the scoring table. The final output class is determined by selecting the class that is maximally non-dominated, and the decision rule is as follows:
(8)$$H=\underset{i=1,\ldots ,N}{\arg\max}\left\{1-\underset{1\leq j\leq N,j\neq i}{\max}l_{ij}^{\prime }\right\}$$In the equation, $l_{ij}^{\prime }$ is the normalized scoring table, and the calculation methods for $l_{ij}^{\prime }$ and $L_{ij}$ are as follows:(9)$$l_{ij}^{\prime }=\left\{\begin{array}{ll} L_{ij}-L_{ji},\; & L_{ij}>L_{ji}\\ 0,\; & \mathrm{other} \end{array}\right.$$
(10)$$L_{ij}=\frac{l_{ij}}{l_{ij}+l_{ji}}$$

### 4.3. The Proposed LVW-MECO Algorithm

#### 4.3.1. M-LVW Algorithm

LVW algorithm [38] uses the error rate of classifiers as the evaluation criterion for feature subset selection. It is unable to adapt to different practical needs by setting different performance evaluation metrics for classifiers and optimizing the feature subset selection based on those metrics. To address this issue, this chapter proposes an improved version of the LVW algorithm called the M-LVW algorithm, as shown in Algorithm 2. With this improvement, the M-LVW algorithm takes the evaluation criterion for feature subsets as an input parameter. This parameter represents a certain performance evaluation metric of the classifier and can be set according to the specific requirements. For example, the evaluation criterion for feature subsets can be set as accuracy, error rate, F1 score, or other metrics based on practical needs.
(11)$$\mathrm{precision}=\frac{\mathrm{samples}\_\mathrm{correctly}}{\mathrm{Total}\_\mathrm{samples}}$$
(12)Error_rate=1−precision
(13)$$F1=\frac{2\times \mathrm{TP}}{\mathrm{Sample}\_\mathrm{count}+\mathrm{TP}-\mathrm{TN}}$$

In the equation, TP represents the number of correctly classified positive samples, and TN represents the number of correctly classified negative samples.

In Algorithm 2, the $\mathrm{CrossValidation}\left(h\left(F^{\prime },D\right),e\right)$ function returns the value of the performance evaluation metric e for the classifier h on dataset D using 10-fold cross-validation. It calculates the value of the performance evaluation metric e for the classifier h on the feature subset $F^{\prime }$. The evaluation metric e can be error rate, accuracy, F1 score, or other metrics.

The M-LVW algorithm is a wrapper-based feature selection method optimized for the final classifier h. This algorithm utilizes a random strategy for the feature subset search and employs a certain performance evaluation metric e of the classifier, such as the error rate, accuracy, or F1 score, as the evaluation criterion for feature subsets. The M-LVW algorithm generates a random feature subset F′ and measures the performance of the classifier using the evaluation metric e. If the performance of the classifier on the feature subset F′ is better than the current best feature subset F*, or if the performance is comparable but the number of features in F′ is fewer, then F′ is assigned to F* and becomes the current best feature subset. The algorithm terminates when it fails to find a better feature subset for T consecutive iterations and outputs the optimal feature subset F* that achieves the best performance evaluation metric e for the classifier. The flowchart of the M-LVW algorithm is shown in Algorithm 2.
**Algorithm 2:** M-LVW feature selection algorithm**Input:** Performance evaluation metric e (evaluation criterion for feature subsets) of the classifier; dataset D; feature set F; classification algorithm h; and stop condition control parameter T**Output:** feature subset F*Process: 1. Initialize: score=CrossValidationhF,D,e; d=F; F*=F; t=02. **If** the value of evaluation metric e for the classifier is positively correlated with the performance of the classifier, **then**3.    p=True4. **else**
5.    p=False6. **end if**7. **while**
t<T
**do**8.    Randomly generate feature subset F′9.    d′=F′10.   score′=CrossValidationhF′,D,e11.   if p==True AND score′>score  OR   p==False AND score′<score  OR   (score′==score AND d′<d)then12.       F*=F′; t=0; score=score′; d=d′13.   **else**
14.      t=t+1
15.   **end if**16. **end while**
**return**$\;\mathrm{feature}\;\mathrm{subset}\;F^{*}$


#### 4.3.2. LVW-MECO Algorithm

In this section, the M-LVW algorithm is further extended to feature selection in the OVO setting, proposing an improved LVW feature selection algorithm called the LVW-MECO algorithm. In this algorithm, each binary classifier is trained with the class that has a larger number of samples as the positive class and the other class as the negative class. The binary classifiers in the OVO setting are also referred to as base classifiers. The LVW-MECO algorithm sets the evaluation criteria for the parameter feature subsets of the M-LVW algorithm as the accuracy and F1 score of the base classifiers in a sequential manner for feature selection. The LVW-MECO algorithm consists of the following two stages:

The wrapper-based feature selection stage uses the accuracy of the binary classifiers as the evaluation criterion for the feature subsets. In the LVW-MECO algorithm, each of the k base classifiers in the OVO setting are applied with the M-LVW algorithm individually. The evaluation criterion for the parameter feature subsets of the M-LVW algorithm is set as the accuracy of the corresponding base classifier. This process selects different feature subsets $F_{i}$ for the k binary classifiers.

The stage of finding optimal feature subsets for r binary classifiers with lower F1 values on the validation set. The LVW-MECO algorithm aims to optimize the accuracy of the multi-classifier composed of binary classifiers on the validation set. It selects better feature subsets for r binary classifiers with lower F1 values on the validation set. The specific procedure is as follows: The LVW-MECO algorithm applies the M-LVW algorithm again to these r binary classifiers one by one. The evaluation criterion for the parameter feature subsets of the M-LVW algorithm is set as the F1 value of the corresponding base classifier. Different feature subsets $F_{i}{}^{\prime }$ are selected for these r binary classifiers. Then, based on the accuracy of the multi-classifier composed of binary classifiers on the validation set, these r binary classifiers select the best feature subset from the two feature subsets F_i_ and $F_{i}{}^{\prime }$ that they have chosen individually.

The LW-MECO algorithm flow is as follows. Step 1 corresponds to the wrapper-based feature selection stage using the accuracy of the binary classifiers as the evaluation criterion for the feature subsets. Steps 2 to 5 correspond to the stage of finding optimal feature subsets for r binary classifiers with lower F1 values on the validation set. The flowchart of the LVW-MECO algorithm is shown in Algorithm 3.
**Algorithm 3:** LVW-MECO feature selection algorithm**Input:** Dataset D containing N classes; feature set F; base classifier (binary classifier) algorithm h in OVO; aggregation strategy s in OVO; stopping condition control parameter T in M-LVW algorithm; and number of binary classifiers to be optimized r**Initialization:** number of binary classifiers k=N(N−1)/2**Step 1:** Apply the M-LVW algorithm to the k binary classifiers in the OVO setting individually and set the evaluation criterion for the parameter feature subsets as the accuracy of the corresponding binary classifier. This process selects k different feature subsets F1,F2,…,Fk for the k binary classifiers h1,h2,…,hk.**Step 2:** By applying the 10-fold cross-validation method, the F1 values corresponding to the k binary classifiers are calculated. All binary classifiers are then sorted in ascending order based on their F1 values.**Step 3:** For j = 1 to r /*Repeat the following Steps 4–5 sequentially for the r binary classifiers with lower F1 values.*/**Step 4:** M-LVW algorithm is used for the pj binary classifier, and the evaluation criterion of its parameter feature subset is set as the F1 value of the binary classifier, and the feature subset Fpj′ is selected for the pj binary classifier, and hpj′ is used as the feature set of the binary classifier e.**Step 5:** The k binary classifiers of h1,h2,…,hk are used to form a multi-classifier H with aggregation strategy s; hpj′ and other binary classifiers hi, i≠pj, and a multi-classifier is formed by aggregation strategy s; then, the accuracy of H and H′ is obtained by using the 10-fold cross-verification method. If the accuracy of H′ is greater than H, then Fpj=Fpj′, hpj=hpj′.**Output:** The subset of features F1,F2,…,Fk corresponding to each binary classifier in OVO 

#### 4.3.3. HSACEC Algorithm

In cases where a significant disparity exists between the sample counts of the majority and minority classes, utilizing a straightforward undersampling approach founded solely on the mean classification error rate within clusters can result in the exclusion of numerous majority class samples. To confront this concern, this study combines the cluster-based undersampling method, reliant on the mean classification error rate within clusters, with the SMOTE (Synthetic Minority Over-sampling Technique). This amalgamation gives rise to the HSACEC (Hybrid Sampling Algorithm for Classifying with Error Cost), which facilitates the attainment of a balanced dataset. This algorithm curtails the quantity of majority class samples through the utilization of the cluster-based undersampling method and simultaneously augments the quantity of minority class samples using the SMOTE method.

Algorithm 4 outlines the workflow of the HSACEC algorithm. The input parameter “m” approximates the count of balanced samples in the resultant balanced dataset “Q” for each class. It is advisable to designate “m” as the median of sample numbers within each class in the original imbalanced dataset. In this study, the class with a sample count surpassing “m” is designated as the majority class, whereas the class with a sample count below “m” is labeled as the minority class. Non-majority classes encompass both the minority class and classes with a sample count equal to “m”. The classifier algorithm “h” within the HSACEC algorithm pertains to the finalized chosen classifier algorithm.
**Algorithm 4:** HSACEC hybrid sampling algorithm**Input:** Dataset D containing class N; equilibrium sampling number m; classifier algorithm h; and the number of fractional samples T.**Output:** Equilibrium sample set Q
1.Set the oversampling rate for each minority class according to Formulas (3) and (4), use SMOTE algorithm to synthesize new samples for each minority class, and add them to dataset D;2.Calculate the single rated sampling quantity z;3.The K-means algorithm is used to cluster the sample sets of each category in dataset D, generate z clusters for each category, and extract the representative points of the clusters from each cluster; a total of N*z samples were extracted and added to the balanced data set Q to realize the initialization of the data set Q, and the sampling times t=1 were counted. Then, remove the extracted samples from dataset D,D=D−Q;4.**for** t=2,3,…,T/*For each sample, repeat steps 5–15.*/ 5.   Training classifier h using balanced data set Q; 6.   **For** every majority class in dataset D is class i 7.        The K-means algorithm is used to cluster the sample set Si of class i in dataset D and generate $\min\left(m,\left|S_{i}\right|\right)$ clusters. 8.        The samples in sample set Si are classified by classifier h, and the average classification error rate $V\left(C\right)$ of the samples in each cluster of Si is calculated by classifier h.; 9.        The $V\left(C\right)$ of all clusters is sorted in descending order, the clusters corresponding to the first $\min\left(z,\left|S_{i}\right|\right)$ with a larger $V\left(C\right)$ value are screened out, and the representative points of these clusters are extracted and added to the balanced dataset Q, and then the extracted samples are removed from the dataset D, D=D−Q; 10.  **end for**
11.  **For** every non-majority class in dataset D is class j 12.     The sample set S of class j in dataset D is clustered by K-means to generate $\min\left(z,\left|S_{j}\right|\right)$ clusters; 13.     The representative points of each cluster are extracted and added to the balanced dataset Q, and the extracted samples are removed from dataset D, D=D−Q; 14.  **end for**15.**end for**16.**return:** equilibrium sample set Q

### 4.4. Network Intrusion Detection Model Based LW-MECO

To improve the accuracy of network intrusion detection models by eliminating redundant and irrelevant features, this chapter applies the LVW-MECO algorithm to construct a network intrusion detection model. A network intrusion detection model based on the LVW-MECO algorithm, called the LVW-MECO-IDM model, is proposed. This model utilizes the OVO decomposition strategy, where the base classifiers are binary classification BP neural networks. The aggregation strategy is based on voting. Figure 2 illustrates the network intrusion detection model based on LVW-MECO, which consists of the following five components: Collect network intrusion data.Preprocess the raw data. First, convert the categorical features of the data into numerical values. Then, perform Z-score normalization on the data.LVW-MECO-based feature selection. Use the LVW-MECO algorithm to select different feature subsets for each base classifier in the OVO setting.OVO-based multiclass classification. Train each base classifier in the OVO setting using the training set. Combine the base classifiers into a multiclass classifier using voting. Use this multiclass classifier to identify intrusion data.Output detection results and respond. Based on the results of network intrusion detection, when intrusion behavior is detected, execute various necessary response measures such as alarms, network disconnection, and other actions.

## 5. Experimental Analysis

### 5.1. Experimental Data

This experiment utilizes the IoT-23 and KDDCUP99_10% datasets. The KDD Cup 99_10% dataset encompasses five distinct categories: Normal, Dos (Denial-of-service), Probe (Surveillance or Probe), U2R (User to Root), and R2L (Remote to Local). The latter four categories fall within the realm of attack types. The quantity of samples within each category is detailed in Table 1. Each individual sample consists of 41 features, which, according to their semantic significance, can be categorized into three groups: fundamental features of network connections, content-related features of connections, and features based on host and temporal traffic.

IoT-23 [18] is a large-scale dataset containing both normal and malicious network traffic in the context of the Internet of Things. It was released by the Stratosphere lab in 2020. The dataset consists of 20 malicious traffic scenarios and 3 normal traffic scenarios. It provides both raw PCAP files and log files based on flow features processed by the Latest Zeek development release tool. For this experiment, we utilize the labeled flow feature log files. The log files contain 18 specific features, excluding the last two columns, which represent the labels. Please refer to Table 2 for the detailed feature descriptions.

### 5.2. Evaluation Criteria

The performance evaluation indicators of the network intrusion detection model used in this chapter include accuracy, detection rate (DR), and false alarm rate (FAR).
(14)$$Detection\_rate=\frac{intrusion\_samples\_detected}{Total\_intrusion\text{\_}samples}$$
(15)$$False\text{\_}alarm\_rate=\frac{Misreport\_samples}{Total\text{\_}normal\_sample}$$

### 5.3. Contrast Model

The comparative models used in this experiment include OBPNN, LVW-OBPNN, F1-LVW-OBPNN, and MFFS-OBPNN models. These comparative models, similar to the LVW-MECO-IDM model, employ the OVO decomposition strategy and use the BP neural network as the base classifier algorithm. The BP neural network structure in these models consists of a single hidden layer. The number of input layer neurons is set to the number of features selected by the respective feature selection algorithm used in each model. The number of neurons in the hidden layer and output layer is the same as that in the BP neural network of the LVW-MECO-IDM model.

The only difference between these comparative models and the LVW-MECO-IDM model proposed in this chapter lies in the adoption of different feature selection algorithms, which select different feature subsets for each base classifier. The LVW-MECO-IDM model utilizes the LVW-MECO feature selection algorithm proposed in this chapter. The feature selection algorithms used by the four comparative models are described as follows: OBPNN model: No feature selection is performed, and this model uses all features.LVW-OBPNN model: Each base classifier adopts the LVW algorithm for feature selection. The error rate of the base classifier serves as the sole evaluation criterion for the feature subset, which is equivalent to using the accuracy [38] of the base classifier as the evaluation criterion for the feature subset. Each base classifier selects a different feature subset.F1-LVW-OBPNN model: Each base classifier adopts an improved version of the LVW algorithm proposed in reference [39] for feature selection. The evaluation criterion for the feature subset in the LVW algorithm is changed to the F1 score of the classifier. Therefore, this model uses the F1 score of the base classifier as the sole evaluation criterion for the feature subset, resulting in different feature subsets selected by each base classifier.MFFS-OBPNN model: Each base classifier adopts the Multi-filter Feature Selection Approach (MFFS) algorithm [40] for feature selection. This algorithm ranks features using filter-based feature selection algorithms based on L1-LR (Logistic Regression), SVM (Support Vector Machine), and RF (Random Forest). Features with rankings below a threshold are removed, and similar features are grouped together. The highest-ranked feature is selected from each cluster, and finally, the features selected by the three feature selection algorithms are combined.

### 5.4. Experimental Parameter Setting

The parameter configurations for the LVW-MECO-IDM model are displayed in Table 3, encompassing chiefly the parameters of the LVW-MECO algorithm and the architecture of the BP neural network. In the One-vs-One (OVO) methodology, all BP neural networks adopt a singular hidden layer configuration. For the i-th BP neural network, the count of input layer neurons corresponds to the number of features contained within the corresponding feature subset Fi, generated by the LVW-MECO algorithm. The hidden layer comprises 15 neurons, while the output layer comprises 2 neurons.

### 5.5. Analysis of Experimental Results

The feature selection results of the LVW-MECO algorithm in the LVW-MECO-IDM model are shown in Table 4. From the table, it can be observed that the original dataset contains a large number of redundant or unimportant features. However, after applying the LVW-MECO algorithm for feature selection, these redundant and unimportant features are effectively eliminated.

Table 5 presents the performance comparison results between the LVW-MECO-IDM model and the four comparative models: OBPNN, LVW-OBPNN, F1-LVW-OBPNN, and MFFS-OBPNN. Compared to the other four models, the LVW-MECO-IDM model proposed in this chapter achieves higher recall rates for the Normal, Dos, Probe, R2L, and U2R categories. In terms of overall performance, the LVW-MECO-IDM model demonstrates a higher accuracy, detection rate, and lower false positive rate. Since the only difference between the other four models and the LVW-MECO-IDM model lies in the adopted feature selection algorithms, it can be observed that the LVW-MECO algorithm used in the LVW-MECO-IDM model outperforms LVW, the improved LVW algorithm proposed in reference, and the MFFS feature selection algorithm in terms of the accuracy, detection rate, and false positive rate.

The presence of numerous redundant and irrelevant features within the dataset diminishes the accuracy of network intrusion detection models. Both the LVW-OBPNN model and the F1-LVW-OBPNN model integrate feature selection algorithms. Specifically, the LVW algorithm employs the base classifier’s error rate as the evaluation criterion for the feature subset, while the enhanced LVW algorithm employs the base classifier’s F1 score as the evaluation criterion for the feature subset. These two algorithms utilize a singular performance evaluation metric of the classifier as the exclusive criterion for evaluating the feature subset. However, this approach falls short of offering a comprehensive assessment of the base classifiers’ performance in the context of feature selection.

The LVW-MECO-IDM model utilizes the LVW-MECO feature selection algorithm proposed in this chapter. The LVW-MECO algorithm employs both accuracy and F1 score as evaluation metrics to assess the performance of the base classifiers from different perspectives for feature selection. This enables the identification of feature subsets that result in improved accuracy for the multi-classifier ensemble composed of the base classifiers.

On the other hand, the MFFS-OBPNN model adopts the MFFS algorithm, which is a filter-based feature selection algorithm. The feature selection process of MFFS is independent of the final classifier to be used. In contrast, the LVW-MECO-IDM model employs the LVW-MECO algorithm, which is a wrapper-based feature selection algorithm. This algorithm directly optimizes the performance of the final classifier by using it as the evaluation criterion for the feature subset. Therefore, the LVW-MECO algorithm can enhance the performance of the final classifier.

In summary, the network intrusion detection model LVW-MECO-IDM, which utilizes the LVW-MECO algorithm, effectively improves classification accuracy, detection rate, and reduces false positives.

From Table 6, it can be observed that after feature selection, the training time of the models significantly decreases. The proposed LVW-MECO-IDM model has a slightly longer training time compared to the LVW-OBPNN model. This indicates that the LVW-MECO feature selection algorithm can to some extent reduce the training time of the classifier. The reason behind this is that the LVW-MECO algorithm effectively reduces the feature dimensionality, leading to improved efficiency in training the classifier.

## 6. Conclusions

This paper introduces methodologies aimed at enhancing the precision of network intrusion detection models through the elimination of redundant and nonessential features. Initially, a hybrid sampling algorithm named HSACEC is proposed, leveraging the average classification error rate within clusters. This algorithm effectively resolves the challenge of discarding a substantial number of majority class samples in scenarios marked by a notable imbalance between majority and minority class samples. Subsequently, an enhanced LVW feature selection algorithm (LVW-MECO) grounded in One-Versus-One (OVO) strategy and multiple evaluation criteria is presented.

The LVW-MECO algorithm is synergistically employed with a BP neural network, culminating in the construction of the network intrusion detection model termed LVW-MECO-IDM. Through empirical validation, the superior performance of the LVW-MECO algorithm is unequivocally demonstrated. The network intrusion detection model, LVW-MECO-IDM, which embraces the LVW-MECO algorithm, delivers remarkable advancements in classification precision, detection rate, and the mitigation of false positives.

The LVW-MECO algorithm presented in this paper belongs to a feature selection method that does not alter the original features. While it performs well when handling independent features, its effectiveness diminishes when dealing with interdependent features. Therefore, our next research endeavor will be focused on addressing feature interdependencies. We intend to explore feature extraction methods to tackle this issue, accompanied by the introduction of quantitative analysis techniques. This approach aims to more precisely and comprehensively extract deeper analyses and insights from existing research findings.

## Figures and Tables

**Figure 1 sensors-23-07434-f001:**
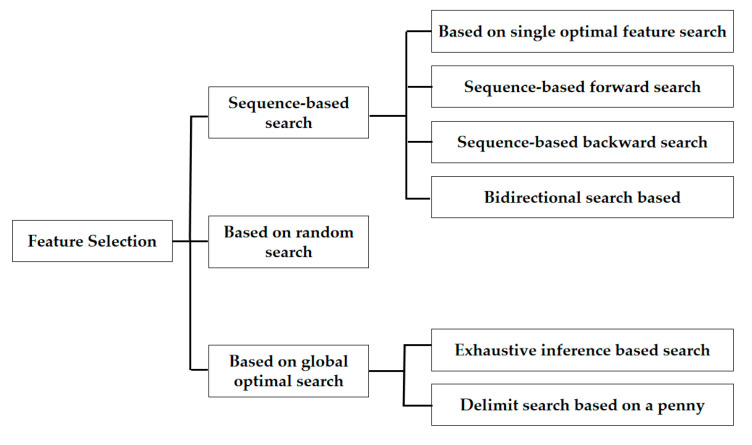
Feature selection method is divided according to the search strategy.

**Figure 2 sensors-23-07434-f002:**
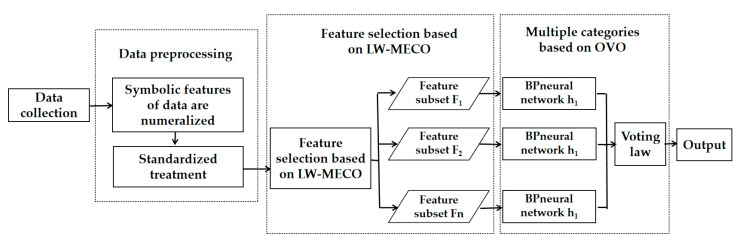
Network intrusion detection model based on LHW-MECO.

**Table 1 sensors-23-07434-t001:** KDD CUP 99_10% dataset.

	Normal	Dos	Probe	R2L	U2R	Unbalance
Training set	uid	391,458	4107	1126	52	7528.04
Test set	id.orig_h	229,853	4166	16,189	228	1008.13

**Table 2 sensors-23-07434-t002:** Feature names and descriptions.

ID	Feature	Description
1	uid	The unique ID of the stream
2	id.orig_h	Source IP address
3	id.orig_p	Source port number
4	id.resp_h	Destination IP address
5	id.resp_p	Destination IP address
6	proto	Agreement
7	service	Dhcp, dns, http, ssh
8	duration	Flow duration
9	orig_bytes	Source sends payload bytes
10	resp_bytes	Destination sends payload bytes
11	conn_state	Connection state
12	local_orig	Source status address flag bit
13	local_resp	Destination status address flag bit
14	missed_bytes	Bytes lost
15	orig_pkts	Number of source address packets
16	orig_ip_bytes	Bytes of the source IP layer
17	resp_pkts	Number of destination packets
18	resp_ip_bytes	Bytes of the destination IP layer

**Table 3 sensors-23-07434-t003:** Parameter setting of LW-MECO-IDM model.

Argument	Value
Data set	IoT-23
OVO’s aggregation strategy s	Voting law
Stop condition control parameter T in M-LVW algorithm	100
The number of binary classifiers to be optimized in the LW-MECO algorithm r	4
Structure of the I-BP neural network in OVO	|F_i_|:15:2

**Table 4 sensors-23-07434-t004:** Feature selection results of the LW-MECO algorithm.

Base Classifier	Classification Category	Feature Selection Result
h_1_	Dos, Normal	2, 4, 5, 6, 8, 9, 10, 11, 13, 14, 15, 16, 17, 18, 19, 20, 21, 23, 25, 27, 29, 31, 32, 33, 34, 35, 36, 37, 38, 39, 40, 41
h_2_	Dos, Probe	3, 7, 8, 12, 23, 25, 27, 28, 29, 30, 32, 33, 34, 36, 38, 39, 40, 41
h_3_	Dos, R2L	1, 23, 31, 39, 40
h_4_	Dos, U2R	3, 8, 10, 12, 14, 17, 23, 25, 26, 28, 31, 32, 34, 35, 40
h_5_	Normal, Probe	2, 3, 7, 8, 12, 23, 24, 25, 26, 28, 32, 33, 34, 36, 38, 39, 40
h_6_	Normal, R2L	2, 3, 7, 11, 13, 14, 16, 18, 22, 24, 26, 27, 32, 34, 35, 36, 40
h_7_	Normal, U2R	2, 3, 10, 11, 12, 14, 17, 18, 23, 24, 25, 26, 31, 32, 33, 35, 36, 38
h_8_	Probe, R2L	2, 4, 5, 6, 7, 8, 11, 13, 15, 16, 17, 18, 19, 20, 21, 22, 23, 24, 25, 27, 28, 29, 30, 31, 33, 34, 36, 37, 38, 39, 40, 41
h_9_	Probe, U2R	2, 3, 14, 26, 30, 33, 40
h_10_	R2L, U2R	2, 3, 5, 10, 12, 16, 17, 18, 19, 22, 23, 29, 30, 32, 33, 34, 37, 39, 40, 41

**Table 5 sensors-23-07434-t005:** Performance comparison of different models.

Model	Recall Rates for All Categories (%)					Precision (%)	Detection (%)	False Alarm (%)
	Normal	Dos	Probe	R2L	U2R			
OBPNN	97.25	97.56	82.37	12.36	13.34	92.80	91.72	2.75
LVW-OBPNN	99.13	99.11	95.17	22.04	18.60	94.99	93.99	0.87
F1-LVW-OBPNN	98.72	98.85	95.06	23.24	20.36	94.78	93.83	1.28
MFFS-OBPNN	98.12	97.94	93.31	20.64	15.62	93.83	92.79	1.88
LVW-MECO-IDM	99.60	99.84	95.32	28.82	20.53	95.98	95.10	0.40

**Table 6 sensors-23-07434-t006:** Model training time after feature selection.

Model	Training Time/s
OBPNN	180.25
LVW-OBPNN	53.37
F1-LVW-OBPNN	59.26
MFFS-OBPNN	57.35
LVW-MECO-IDM	55.34

## Data Availability

Data openly available in a public repository.

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
